# Saffron against Neuro-Cognitive Disorders: An Overview of Its Main Bioactive Compounds, Their Metabolic Fate and Potential Mechanisms of Neurological Protection

**DOI:** 10.3390/nu14245368

**Published:** 2022-12-17

**Authors:** Débora Cerdá-Bernad, Leonor Costa, Ana Teresa Serra, Maria Rosário Bronze, Estefanía Valero-Cases, Francisca Pérez-Llamas, María Emilia Candela, Marino B. Arnao, Francisco Tomás Barberán, Rocío García Villalba, María-Teresa García-Conesa, María-José Frutos

**Affiliations:** 1Research Group on Quality and Safety, Agro-Food Technology Department, CIAGRO-UMH, Centro de Investigación e Innovación Agroalimentaria y Agroambiental, Miguel Hernández University, 03312 Orihuela, Spain; 2iBET, Instituto de Biologia Experimental e Tecnológica, Apartado 12, 2781-901 Oeiras, Portugal; 3Instituto de Tecnologia Química e Biológica António Xavier, Universidade Nova de Lisboa, Av. da República, 2780-157 Oeiras, Portugal; 4iMED, Faculdade de Farmácia da Universidade de Lisboa, Av. das Forças Armadas, 1649-019 Lisboa, Portugal; 5Department of Physiology, Faculty of Biology, University of Murcia, 30100 Murcia, Spain; 6Department of Plant Biology (Plant Physiology), Faculty of Biology, University of Murcia, 30100 Murcia, Spain; 7Research Group on Quality, Safety and Bioactivity of Plant Foods, Centro de Edafología y Biología Aplicada del Segura (CEBAS), Spanish National Research Council (CSIC), Campus de Espinardo, 30100 Murcia, Spain

**Keywords:** apocarotenoids, bioavailability, crocin, crocetin, colon microbiota, neurological disorders, flavonoids, plant by-products, saffron

## Abstract

Saffron (*Crocus sativus* L.) is a spice used worldwide as a colouring and flavouring agent. Saffron is also a source of multiple bioactive constituents with potential health benefits. Notably, saffron displays consistent beneficial effects against a range of human neurological disorders (depression, anxiety, sleeping alterations). However, the specific compounds and biological mechanisms by which this protection may be achieved have not yet been elucidated. In this review, we have gathered the most updated evidence of the neurological benefits of saffron, as well as the current knowledge on the main saffron constituents, their bioavailability and the potential biological routes and postulated mechanisms by which the beneficial protective effect may occur. Our aim was to provide an overview of the neuroprotective effects attributed to this product and its main bioactive compounds and to highlight the main research gaps that need to be further pursued to achieve full evidence and understanding of the benefits of saffron. Overall, improved clinical trials and adequately designed pre-clinical studies are needed to support the evidence of saffron and of its main bioactive components (e.g., crocin, crocetin) as a therapeutic product to combat neurological disorders.

## 1. Saffron and Its Constituents

The culinary spice of saffron is widely used as an ingredient in the preparation of many foods around the world due to its distinctive aroma, taste, and color. It is also used in the cosmetic and textile-dying industry [1]. This spice is prepared from the dried stigmas of the plant *Crocus sativus* L. For the production, the red stigmas are cautiously hand-separated from the rest of the flower. Overall, a large number of flowers (≈60–80 Kg) are needed to obtain ≈1 Kg of dried saffron, and thus, this process generates a large quantity of waste product (≈93% of the flower mass) which is mostly constituted by tepals (≈78%) [2] (Figure 1).

Saffron is also used for therapeutic purposes since it contains many constituents with multiple health benefits. The dried saffron spice (stigmas) contains primarily carbohydrates (≈60%), fiber (≈14%), proteins (≈14%), lipids (≈9%), and minerals (≈7%; mostly K, P, Mg, Ca, Fe, Na). They also contain some soluble sugars, and small quantities of mucilage, vitamins (mainly thiamine and riboflavin), amino acids, and organic acids [3,4]. In addition, the saffron stigmas contain hundreds of volatile and non-volatile secondary metabolites, some of which are also categorized as bioactive compounds due to their chemical and biological properties. The main bioactive compounds of the saffron stigmas are carotenoid-related compounds such as crocetin, crocin, picocrocrocin, and safranal [5] (Figure 2). The quality and functional activity of this valuable product has been attributed to these compounds that can be found at highly variable levels depending on different environmental and agronomic factors [6]. β-Carotene and zeaxanthin (a xanthophyll) found in the stigmas of *C. sativus* are the precursors of crocetin and crocin [7]. Crocins are the water-soluble carotenoids responsible for the yellow-orange color of saffron and are made of crocetin (8,8′-diapocarotene-8,8′-dioic acid) in which both of the carboxylic groups have been esterified with a glycoside moiety. Crocins are found as a mixture of *trans*- and *cis-*isomers of which the most abundant are the *trans*-crocetin di-(β-D-gentiobiosyl ester) (*trans*-crocin-1 or *trans-*4-GG), and *trans*-crocetin (β-D-glucosil)-(β-D-gentiobiosil) (*trans-*3-Gg) [2,8]. The levels of crocins can vary from ≈5 to 40% of the dry weight of the processed stigmas [6]. The principal volatile compounds of saffron are terpenoids, mainly terpenic alcohols and their esters [1,9]. Picrocrocin is a monoterpene glycoside responsible for the bitter flavor of saffron (Figure 2). This compound can serve as a marker to identify true saffron because it has only been characterized in the genus *Crocus* sp. [2]. Safranal, not initially present in the fresh stigmas, is formed during their drying and storage process by the oxidative and/or enzymatic degradation of picrocrocin. Safranal is an aldehyde monoterpene responsible for the predominant aroma of saffron, accounting for up to 70% of the volatile compounds. Many other volatile molecules have been identified and associated with the aroma of saffron such as ketones (isophorone-related compounds, cyclopentanone), aldehydes (nonanal), esters (furanone), C13-norisoprenoids, linear saturated hydrocarbons, acetic acid, etc. [10,11]. Another important group of bioactive compounds identified and quantified in the saffron stigmas are the flavonoids, i.e., flavonols (mostly kaempferol and quercetin), flavanols (epicatechin), anthocyanins (malvidine), as well as small quantities of up to 20 different phenolic compounds (e.g., gallic acid, vanillic acid, rosmarinic acid, chlorogenic acid, etc.) (Figure 2) and of hydrolyzable and condensed tannins [6]. The structure of some of the most representative bioactive compounds of saffron is shown in Figure 2 [12,13,14,15,16,17,18,19].

The nutritional composition of the saffron tepals has also been reported, i.e., carbohydrates (≈65%), proteins (≈8%), lipids (≈2%), fiber (≈28%), minerals (≈6%; mostly K, Ca, P, Na, Fe, Mg, Zn, Cu), and residual moister. In addition, the soluble sugars [3] and vitamin C have also been identified in the saffron tepals [4]. Regarding crocins and crocetin, only small quantities of *trans*-crocetin and *trans-*crocin (*trans-*4-GG and *trans*-3-Gg) have been detected in the floral by-products [6,11]. Other carotenoids such as lutein in the form of diesters (with stearic, myristic, palmitic, and lauric fatty acids) have also been identified in the flowers [20]. The flower by-product also contains substantial quantities of bioactive molecules such as flavonoids (anthocyanins, flavonols), which are responsible for the color of the tepals, as well as phenolic acids and tannins. The main identified and quantified anthocyanins were delphinidin, petunidin, malvidin, cyanidin, and peonidin [6]. The quantities of these compounds are variable and, for example, delphinidin 3,5-di-O-β-glucoside was found to be very abundant in the plant perigonium [2,21]. As for the flavonols, quercetin, myricetin and kaempferol are also present in the dry-flowers at variable quantities [6]. Kaempferol has been reported to be one of the most abundant compounds accounting for up to ≈84.0% of the total content present as kaempferol 3-O-sophorol (55.4%), followed by kaempferol 3-O-glucoside and kaempferol 3-O-sophoro-7-O-glucoside [2,20]. Isorhamnetin glycosides and kaempferol have also been found in the saffron pollen, while quercetin (3, 3′, 4′, 5, 7-pentahydroxyflavone), as well as some flavones, have been detected in the saffron tepals and in the leaves [5]. Other bioactive compounds also identified in the dried extracts from the flowers of saffron are phenolic compounds and tannins. The most abundant phenolics reported in the dried flowers include gallic acid, 4-hydroxy-benzoic acid, salicylic acid, chlorogenic acid, sinapic acid, coumaric acid, or rosmarinic acid [4,6]. Several volatile compounds have also been recently identified in the fresh saffron flowers, i.e., 3-hydroxy-2-butanone (acetoin), butanoic acid, acetic acid, butanal, carbolic acid, phenylethyl alcohol, among others [11]. After applying different drying procedures (air-oven or freeze-drying) several volatile compounds were still present in the dried flowers, including 2,3-butanediol; 2-furancarboxaldehyde 5-methyl-; ethanone, 1-(1H-pyrrole-2-yl)-; 4H-pyran-4-one, 2,3-dihydro-3,5-dihydroxy-6-methyl-(DDMP); 2-cyclohexen-1-one, 2-methyl-5-(1-methylethyl)-, (S)-; 2-cyclohexen-1-one, 2-methyl-5-(1-methylethyl)- [11]. A summary of the main compounds identified in the saffron stigmas and waste flower products (mostly tepals) is listed in Table 1.

The application of more powerful analytical techniques continues identifying an increasing number of metabolites in the stigmas and petals of saffron. In a very recent report using targeted metabolomics, up to 800 molecules were identified and classified into 35 classes in the stigmas and petals of the Chinese saffron [22]. The results confirm a similar chemically qualitative composition between the two parts of the plant but with highly variable and differential quantitative composition. Overall, the amounts of crocins, picrocrocin, safranal, and crocetin, are mainly present in the stigmas but the flower by-product (mainly tepals) also constitute a source of a great variety of other bioactive compounds such as flavonoids and phenolic acids with potential industrial and health applications.

## 2. Neuro-Cognitive Protective Effects of Saffron in Humans

The therapeutic applications of saffron have been recurrently evaluated in many human clinical trials, as well as thoroughly revised in several reviews and meta-analyses. Overall, the supplementation with this product and (or) with some of their main bioactive constituents has been associated with the improvement of a wide range of diseases including cardiovascular and metabolic disorders, ocular diseases, urogenital and sexual disorders, inflammatory and immune-related diseases, and neurological disorders. One of the most extensively investigated therapeutic effects of saffron and of some of its major bioactive constituents (crocins/crocetin), is the protective activity against a number of neurocognitive disorders including Alzheimer’s disease (AD), depressive and anxiety disorders, obsessive-compulsive disorders, attention deficit, sleeping disorders and (or) cognitive function impairment [23].

The main aim of this section was to critically revise and update the evidence of the neuroprotective effects of saffron in a selection of human clinical trials that have evaluated the effectiveness of saffron or of some of its main components against neurocognitive disorders. We examined a total of 23 double-blind randomized clinical trials (RCTs) carried out between 2010 and 2022 (Appendix A) [24,25,26,27,28,29,30,31,32,33,34,35,36,37,38,39,40,41,42,43,44,45]. Overall, these studies were conducted in patients with signs of depression, anxiety, sleeping problems, or cognitive disorders. The average sample size was ≈50–60 participants divided into two parallel groups, i.e., the treated group (saffron, mostly provided as an extract in capsules) and the placebo (PLA) group (PLA capsules) or a comparative drug (drug-containing capsules). The administered doses were typically ≈30 mg/day except for a few studies where the dose was increased up to 50–100 mg/day [26,27,38], and the duration of the treatment was most frequently between 6 and 8 weeks although a few interventions lasted for longer (12 weeks) [28,30,38] or even one year [43,44,45]. Some studies reported the presence and quantity of some of the major bioactive compounds (crocin, safranal) in the saffron extract [29,30,32,33,34,35,36,40,44,45] but the full composition of the extracts was not indicated. A few studies were conducted with the single saffron compounds, crocin or crocetin [25,37,41] but the purity, solubility, and stability were not reported. Regarding the measuring instruments, most studies applied different well-established and validated subjective questionnaires and scoring systems to evaluate the efficacy of the saffron treatment on depression, anxiety, sleeping disorders, quality of life, and cognitive disorders. Only a few studies implemented some additional objective measures of sleep quality [40,41].

With independence of the questionnaire employed, most studies reported a significant improvement of the scores measuring depression and anxiety disorders in the group treated with saffron. Several studies that used the Beck depression inventory (BDI) scoring scale reported reductions between −5.0 and −11.6 points [25,26,28,31,35,37,38]. Three other studies applied the Hamilton depression rating score (HDRS) system and reported reductions between −7.1 and −11.2 points [27,29,33]. The score for anxiety was also reduced between −1.7 and −12.0 points depending on the study and system used [25,30,32,34,38]. Overall, when the change in the saffron group was compared with that occurring in the PLA group (true effect), in most cases the difference between groups was significant supporting that the saffron was able to reduce the depression scoring by ≈25–60% and the anxiety scoring by ≈40%. In addition, the differences between saffron and various drugs (sertraline, fluoxetine, or citalopram) were not significant [27,33,34] further reinforcing the beneficial effect of saffron on depression and anxiety. Changes in cognitive impairment were also investigated in five studies [30,42,43,44,45] where saffron also showed relevant efficacy in ameliorating the problem with reductions in the attention deficit and hyperactivity disorder (ADHD), the severe cognitive impairment rating scale (SCIRS), the Alzheimer’s disease assessment scale–cognitive subscale (ADAS-cog) and the clinical dementia rating scale–sum of boxes (CDR-SB) scores [42,44,45], and increasing the results of the mini mental status examination (MMSE) score [30,43]. The reported % of changes were generally lower than those reported for depression, but still significantly different from the PLA tablets [30,43,45] and non-significant against some drugs (methylphenidate, memantine) [42,44] corroborating some cognitive benefits for saffron. We additionally examined a few studies looking at the effects of saffron against sleeping disorders with less clear results and some improvement only in specific subscales or subcomponents of the global sleep quality measurement instruments [26,36,40,41].

Our analysis corroborates a consistent and significant improvement of depression, anxiety, and cognitive impairment associated with the daily intake of moderate quantities of saffron extracts containing bioactive compounds such as crocin, and/or crocetin. The effects seem to be comparable to those of specific pharmacological treatment and thus, the regular intake of this product may contribute to the moderation and control of some of those neurological disorders. Further, saffron also appears to be generally well tolerated with no major adverse effects associated with its daily consumption [46]. To further understand and support the neurological beneficial effects of saffron, it remains essential that we fully elucidate the main bioactive constituents responsible for these effects as well as their bioavailability and potential mechanisms of action. The next sections of this review present a thorough and updated revision of the current knowledge on (i) the absorption, tissue distribution, metabolism, and excretion (ADME) of some of the major constituents and derived metabolites from saffron, and (ii) the potential physiological routes and molecular mechanisms of action of this product.

## 3. Bioavailability of the Main Saffron Bioactive Constituents

### 3.1. Host Metabolism

The bioavailability of the major bioactive compounds present in the stigmas of saffron (crocin, crocetin, safranal, pirocrocin) has not yet been entirely deciphered. Nevertheless, several pre-clinical studies and human intervention studies have already pointed to the first metabolic steps of some of these compounds. A list of animal studies investigating the absorption and metabolism of some of the main saffron compounds is presented in Table 2. Asai et al. [47] first reported in mice the absorption of oral doses of crocin (6.1 mg/Kg) and crocetin (2.1 mg/Kg) derived from the fruits of gardenia (*Gardenia jasminoides*). The results from this study already evidenced a rapid absorption of crocetin into the blood circulation which was first detected in plasma in its intact free form and, later on, as the crocetin mono-glucuronide and di-glucuronide conjugates. These metabolites were also detected in the plasma of mice fed with the crocin. However, the intact forms of crocin (glycosides) were not found, supporting that de-glycosylation occurred as a first step during the intestinal absorption of this compound [47]. A second study conducted in Sprague-Dawley rats corroborated that after oral administration of a high dose of crocin (40 mg/Kg), this compound was still not detected in the plasma of the animals [48]. In this study, crocetin was reported to have a short plasma half-life (≈1 h), even after repeated oral doses, being rapidly eliminated without apparent accumulation in the body. Additional analyses of the 24 h urine and fecal samples revealed that crocin was not detected in the urine, but it was largely excreted in the feces (≈59.5 ± 13.6% after a single oral dose) indicating that this compound reached the large intestine in its intact form [48]. More recently, some dose-response studies conducted by Zhang et al. [49] and by Shakya et al. [50] in Sprague-Dawley rats further confirmed the absorption of crocetin but also evidenced the presence of small quantities of crocin in plasma. In particular, Zhang et al. [49] demonstrated that following the oral administration of crocin, the plasma concentration of this compound was 10 to 50 times lower than that of crocetin. The poorer bioavailability of the crocins was attributed to the polar character of these molecules which reduces their capability to permeate through biological membranes in comparison with that of crocetin. In support of the passive transcellular diffusion of crocetin, in vitro cell studies have shown that the de-glycosylated compound was able to permeate through Caco-2 cell monolayers [51]. Overall, the animal studies collected in Table 2 support that upon de-glyclosylation, crocins are converted into its non-glycosylated C20-dicarboxylic acids, mostly *trans*-crocetin, which is the main compound absorbed. This molecule and (or) its glucuronides have the potential to be distributed across the body and enter into different cells and tissues and thus, crocetin has been proposed as one of the main bioactive compounds of saffron potentially involved in the health benefits of this product, including its neurological benefits [47,49,52].

Up to date, only two human intervention studies have investigated the bioavailability of the bioactive compounds present in saffron (Table 3). In the study by Chryssanthi et al. [53], four healthy volunteers were given a saffron infusion (200 mg of saffron stigmas in 150 mL of hot water) containing mainly crocins (*trans*-crocin-4, *trans*-crocin-3, *cis*-crocin-4, *trans*-crocin-2, *cis*-crocin-3, and *cis*-crocin-1). The authors confirmed that during the infusion preparation, the crocins remained stable and no isomerization or hydrolyzation occurred. After 2 h of the intake, the authors primarily detected crocetin in the plasma of the volunteers (levels ranging from 0.41 to 1.21 µg/mL). Crocetin was still detected at trace quantities (0.03–0.08 µg/mL) even 24 h after the intake of the saffron tea. In a more recent study conducted by Almodóvar et al. [54], the volunteers consumed two different doses (56 mg or 84 mg) of affron^®^ (Pharmactive Biotech Products S. L., Madrid, Spain) a commercial galenic formulation prepared from a saffron extract, containing safranal (0.04% ± 0.01), picrocrocin (3.21% ± 0.07), kaempferol diglucoside (0.13% ± 0.01), crocins (3.63% ± 0.05) and crocetin (0.03% ± 0.01) and several excipients (hypromellose, macrogol (PEG) 8000 and carnauba wax). The results confirmed that after taking the affron^®^ tablets, the main compound detected in plasma was crocetin with the maximum concentrations achieved at 60 min (0.26 ± 0.12 µg/mL) and 90 min (0.39 ± 0.10 µg/mL). Crocin, picrocrocin, or safranal were not detected. In this study, the authors also carried out an in vitro digestion of the affron^®^ extract. They found a significant decrease in the concentration of the crocin isomers (especially *trans*-crocin-4) during gastric and intestinal digestion. During the gastric stage (pH 2, 37 °C, pepsin, 90 min) there was a fast metabolism of crocin into crocetin. Under the conditions of the three phases of the digestion (salivary, gastric, and intestinal digestion), the concentration of safranal was found to be increased by ≈2-fold, whereas picrocrocin decreased by almost 19%. Since safranal is a derivative of picrocrocin, these results support the conversion of picrocrocin to safranal during the digestion [54]. The results of the human studies further confirm a quick hydrolysis of crocin to crocetin (within 1–2h) before going into the bloodstream and that crocetin is the major compound detected and quantified in human plasma. It has been hypothesized that the circulating crocetin may bind to human serum albumin (HAS) as well as to other serum proteins [55,56]. Different techniques (UV–vis absorption, fluorescence quenching, circular dichroism spectroscopies, and molecular docking) have been implemented to confirm the formation of stable complexes between crocetin and HAS that could solubilize and then distribute them through the blood [56].

Concerning the body distribution of the main saffron metabolites, there are not yet many studies detecting and quantifying the presence of any of the main saffron bioactive compounds and (or) their metabolites and conjugates in different tissues. To date, there is only one study conducted by Christodoulou et al. [57] who based on animal experiments, built a computational pharmacokinetic model (in silico) for compartmental and non-compartmental analysis, to predict and calculate the distribution of saffron metabolites in the serum and in specific tissues. They concluded that crocetin could be highly distributed through the body, being transformed into its conjugated glucuronides and then eliminated through the liver and kidneys by enterohepatic recycling. In the heart and lungs, glucuronidation was thought to be potentially faster since no measurable levels of crocetin were detected by the authors. The estimated serum concentrations of crocetin and its glucuronides were not superior to 4.0 µg/mL and 2.5 µg/mL, respectively. In the different tissues, the concentrations ranged from ≈0.5–3.0 µg/mL [57]. Regarding the brain, and in potential connection with the regulatory neurological benefits of saffron, there are no yet in vivo studies evidencing the presence of crocin, crocetin, and (or) its glucuronides in the brain tissues. Nevertheless, in the in vitro study conducted by Lautenschläger et al. [51], it was demonstrated that *trans*-crocetin (27–50 µM) was able to permeate through two types of in vitro models that mimic the blood-brain barrier (BBB), namely primary brain capillary endothelial cells (BCEC) and epithelial cells of the Choroid plexus of the blood-cerebrospinal fluid barrier (BCSFB) isolated from fresh porcine brain. Given the permeation coefficients obtained (1.5 × 10^−6^ ± 0.1 × 10^−6^ cm/s and 3.9 × 10^−6^ ± 0.2 × 10^−6^ cm/s for BCEC and BCSFB, respectively), it was hypothesized that crocetin may be able to penetrate the BBB and reach the central nervous system (CNS) in a quite slow process.

### 3.2. Microbial Metabolism of Saffron

In relationship with the metabolism of the saffron compounds, and, in particular, with the metabolism of crocin and crocetin, there are still important unresolved questions such as the contribution of the intestinal microbiota to this process [51,56,58,59]. A considerable proportion of these compounds can reach the large intestine where they can be subjected to microbial metabolic activity. The study reported by Zhang et al. [60] supports the important role of the gut microbiota in the metabolism of crocin and crocetin. Since none of these compounds have been detected in the brain tissue after oral or intravenous administration, it is hypothesized that further metabolism and (or) interaction with the gut microbiota may yield other not yet identified metabolites that may be implicated in the neurological benefits of these compounds. The role of microbiota was further supported by a pharmacokinetics study where following an oral dose of crocin, a lower amount of crocetin was found in the plasma of pseudo-germ-free rats than in the plasma of normal animals. These results suggest further de-glycosylation of crocin to crocetin by the microbial enzymes and were supported by in vitro studies [60]. In a more recent study, crocin (600 mg/kg) was orally administered to male rats pre-treated with a mixture of antibiotics (cefadroxil, oxytetracycline, erythromycin, for three days). Since the amount of crocetin produced from crocin was higher in the control rats than in the rats pre-treated with antibiotics, the authors concluded that the biotransformation of crocin into crocetin by the intestinal microbiota was a critical step for absorption [50]. Overall, the studies presented here evidence that the main identified bioactive compound in the saffron stigma that is absorbed and metabolized and has the potential to reach the different tissues including the brain is crocetin and its glucuronides. Crocin and crocetin can also reach the large intestine where they may suffer further metabolism by the microbiota.

## 4. Potential Mechanisms of Action Underlying the Neurological Responses of Saffron

As already stated, saffron has been consistently shown to have significant benefits against neurological disorders but, the potential physiological routes and mechanisms by which these effects may occur have not yet been fully elucidated. In the next sections, we gathered a series of studies that were conducted to try to envisage some of the molecular means by which the saffron compounds may exert some neuroprotective action.

### 4.1. In Vitro Cell Studies Supporting the Neurological Effects of the Main Saffron Compounds

One generally accepted hypothesis is that the neuroprotection of saffron may be attributed to its main bioactive compounds and their associated biological properties (i.e., anti-inflammatory, antioxidant, anti-apoptotic activity, etc.) [61]. The saffron compounds (or their derived metabolites) may interact with specific target cells triggering the molecular responses that promote the protective effect. Table 4 includes a series of in vitro studies looking at the effects of crocin, crocetin, and safranal in different cultured cells (from rats or mice) that represent cells of the nervous system such as the microglia (brain macrophages involved in the maintenance of neurons and in injury repair), neuroblastic cells (isolated from an adrenal medulla that constitute undifferentiated precursors of the CNS), or dopaminergic cells (neurons of the CNS that synthesize the neurotransmitter dopamine). These cells can be exposed to neurotoxins, oxidative stress, and (or) other toxic compounds to simulate and investigate brain disorders like ischemia or Parkinson’s disease (PD) [62,63,64,65,66,67,68,69,70]. Overall, these studies support that the potential neuroprotective benefits of the main saffron bioactive compounds may be mediated by (1) anti-inflammatory effects such as the inhibition of inflammatory associated genes and cytokines [63]; (2) antioxidant effects such as the reduction of oxidative stress and reactive oxygen species (ROS) [64,65,66]; and (or) (3) anti-apoptotic effects and improvement of the mitochondrial function [63,65,66,67]. In a more recent study, the anti-apoptotic and antioxidant effects of crocin were also confirmed in a cell model of hippocampus neuronal cells subjected to experimental ischemia. In this model, crocin was shown to modulate the levels of several proteins involved in the autophagy [62].

A-Synuclein (α-Syn) is a presynaptic neuronal protein predominantly expressed in the brain and the main component of the Lewy bodies. Abnormal aggregation and accumulation of this protein have been associated with neurodegenerative diseases such as dementia or PD [71]. Some in vitro studies have looked at the regulatory effects of crocin, crocetin, and safranal on the aggregation/dissociation of α-Syn [68,69]. Inoue et al. [68] reported that the saffron compounds inhibited the aggregation and promoted the dissociation of a solution of α-Syn fibrils in a dose-dependent manner, with crocetin being the most potent compound. The E46K mutant that creates more pathogenic α-Syn fibrils in vitro [72] can also be inhibited by crocin [69]. More recently, the study conducted by Inoue et al. [70] in human neuronal cells further corroborated the protection of these cells against the toxicity generated by the addition of α-Syn to the cell culture medium.

### 4.2. The Effects of Saffron and Its Main Compounds in Animal Models of Neurological Diseases

Table 5 presents the results of recent studies looking at the neurological protection of saffron or its main bioactive compounds (crocin, crocetin, safranal) in animal models [63,70,73,74,75,76,77,78,79,80]. Most of these studies were conducted in rodents (mice or rats) with induced alterations mimicking some of the symptoms of nervous and brain diseases (e.g., hypoxia-ischemia, PD, AD, schizophrenia). Only the study by Inoue et al. [70] investigated the effects of oral saffron and of crocetin in a PD model of *Drosophila* expressing several mutations of human α-Syn in their neurons or eye cells. The results of this study indicated that the intake of saffron (as well as crocetin) protected the neurons and eyes of the flies against the altered aggregation of α-Syn and improved the activity and life expansion of these animals. The study by Abdel-Rahman et al. [73] conducted in a rat model of brain ischemic injury showed that the intraperitoneal injection (i.p.) of high doses of saffron during 3 weeks also exerted some antioxidant and anti-apoptotic responses in the brain cells which could be mediated by the regulation of different proteins including the vascular endothelial growth factor (VEGF).

As the main bioactive component present in the saffron stigma, pure crocin has been repeatedly tested for its neuroprotective effects in different animal models. In a very recent study, the administration of crocin (i.p.; 40 mg/Kg b.w. (body weight) was reported to protect against intracerebral hemorrhage in mice [74] by reducing brain edema and improving a neurological deficient score. These results appeared to be achieved through the regulation of antioxidant activity, the inhibition of ferroptosis in the neurons, molecular regulation of the expression of several enzymes, and the translocation of the nuclear factor erythroid 2-related factor 2 (Nrf2). Crocin was also beneficial against induced hypoxic-ischemic encephalopathy in mice by reducing the brain levels of malondialdehyde (MDA), nitric oxide (NO), and ROS and inhibiting the expression of pro-inflammatory mediators [75].

Alzheimer’s disease (AD) is a neurodegenerative disorder characterized by abnormal tau protein metabolism, the presence of the insoluble extracellular amyloid-β peptide (Aβ) aggregates, an inflammatory response, and cholinergic and free radical damage. At present, there are not many drugs to treat this disease, and new therapies and natural compounds are being investigated [81]. Recently, Hadipour et al. [76] reported that crocin displayed some neuro-modulatory effects against a rat model of AD. The authors showed that the i.p. administration of crocin (30 mg/kg b.w., 12 days) ameliorated the response to the injection of Aβ1-42 into the rats’ frontal cortex. Crocin increased the number of live cells in the hippocampus pyramidal neurons in the CA3 and granular cells in the dentate gyrus regions as well as reversed the loss of axonal, spine, and dendrites arborization in the frontal cortex and CA3 region. Overall, crocin improved synaptic loss and reduced neuronal cell apoptosis. Similar doses of crocin (i.p., 25 or 50 mg/Kg b.w.) also exerted neuroprotective effects in a rat model of MK-801-elicited schizophrenia by modulating the expression of silent information regulator-1 (SIRT1) and brain-derived neurotrophic factor (BDNF). Further, crocin relieved oxidative stress by reducing the levels of MDA and increasing the activity of the antioxidant enzymes catalase (CAT), glutathione peroxidase (GPx), and superoxide dismutase (SOD) in the hippocampus [77]. The antioxidant and mitochondrial function benefits of crocin were also demonstrated by Rao et al. [78] in a mice model of PD. Crocin (i.p., 25 mg/Kg b.w., 7 days) restored the levels of dopamine and α-Syn and enhanced the antioxidant status and mitochondrial enzyme dysfunctions in the striatum of the animal’s brain. Whether all these neuroprotective effects are directly mediated by the circulating crocin (following i.p. injection) remains to be investigated. Based on a metabolomics approach, Karkoula et al. [80] explored the composition of plasma samples after the i.p. administration of *trans*-crocin-4 in C57BL/6 J mice. The results showed that the metabolome changed substantially after the treatment and that some of the identified metabolites played important roles in neuroprotection (i.e., 21-hydroxypregnenolone and 17α-epiestriol involved in the steroid biosynthesis pathway; glutaryl carnitine and riboflavin related to oxidative stress, and betaine involved in inhibitory neurotransmitter production pathways).

Additional studies have investigated the neurological effects of other saffron compounds. Dong et al. [63] reported that the oral administration of crocetin (50 and 100 mg/Kg b.w.) to a C57BL/6 mice model of induced PD attenuated the expression of the tyrosine hydroxylase (TH) marker in the striatal regions, suggesting a protective effect of the dopaminergic neurons. The neuroprotective effects of safranal (i.p., 72.5 and 145 mg/Kg b.w.) were also investigated in adult male rats with transient focal cerebral ischemia induced by middle cerebral artery occlusion (MCAO). Safranal inhibited oxidative stress in the brain tissues by reducing the levels of MDA and increasing the total sulfhydryl (-SH) content [79].

Overall, a variety of pre-clinical studies support the neuroprotective effects of saffron, principally of its major component crocin. These effects may be mediated by triggering a range of antioxidant and anti-inflammatory responses mediated by the molecular regulation of key proteins involved in specifically related pathways [82].

### 4.3. The Role of the Interaction with the Microbiota in the Neurological Effects of Saffron

The gut microbiota has modulatory effects on brain function and human behavior through the bidirectional communication brain-gut axis. Recent evidence suggests that an imbalance in the equilibrium of the gut microbiome known as dysbiosis (alteration of the normal microbiome composition and function) is associated with the development of neurological disorders such as PD, AD, schizophrenia, and depression. It has been proposed that future therapies against these disorders may involve the restoration of the normal microbiome composition and its communication with the host metabolic, immunological, and neurological responses [83]. The dietary intake of bioactive compounds has been shown to modify the growth and distribution of specific bacterial communities towards a healthier gut microbiome and metabolome and may contribute to the attenuation of neurological disorders such as depression, AD or PD by reducing the oxidative and inflammatory processes, as well as the neurotransmitter dysfunction, and neuronal death associated with these diseases [84]. Crocin and crocetin, the main bioactive compounds of saffron, have recently been investigated for their potential ability to modify the gut microbiota composition in relationship with neurological disorders. In particular, the oral administration of crocin-1 for 6 weeks to a mice model of induced depression alleviated gut dysbiosis represented by a decrease in Proteobacteria and Bacteroidetes, *Sutterella* spp. and *Ruminococcus* spp., and an increase in the population of Firmicutes, *Lactobacillus* spp. and *Bacteroides* spp. Crocin-1 also caused a general increase in the α-diversity of the colon microbes and reversed the decreased levels of fecal short-chain fatty acids (SCFAs) observed in the depressed mice [85]. In a second study, crocin-1 was evaluated for its efficacy to regulate lipid metabolism and gut microbiota in a mouse model of chronic corticosterone treatment which can cause microbiota dysbiosis and neurological impairment [86]. At the dose of 40 mg/Kg, crocin-1 was able to increase the α-diversity of the caecum microbiota with a significant decrease in Firmicutes and a decrease in Bacteroidetes. Crocin-1 also alleviated the hepatic lipid disorder. More recently, crocetin was also shown to ameliorate the effects of stress-associated brain damage in association with changes in the gut microbiota in a mice model of chronic restraint stress [87]. In this study, oral administration of crocetin (20, 40, 80 mg/Kg) improved the depressive-like behavior of the stressed animals as well as the hippocampus tissue injuries and the levels of several associated biomarkers (i.e., mitogen-activated protein kinase phosphatase-1 (MKP-1), precursor of brain-derived neurotrophic factor (pro-BDNF), extracellular signal-regulated kinase 1/2 (ERK1/2), phosphorylated cAMP response element-binding (CREB), serum dopamine (DA). In addition, crocetin partially recovered some of the gut microbiota groups altered under the stress conditions. The effects of crocetin were comparable to those of fluoxetine (a common antidepressant drug). The authors concluded that crocetin ameliorated the effects of stress-associated brain damage by regulating the MKP-1-ERK1/2-CREB signaling and the intestinal ecosystem.

Other recent studies have also shown that the supplementation with saffron can modify the gut microbiota composition in association with other anti-inflammatory and metabolic beneficial effects attributed to this product. In the study conducted by Li et al. [88], a Chinese plant extract containing saffron was able to improve several diabetic parameters (e.g., glucose and insulin levels) and hepatic metabolism via the Toll-like receptor (TLR) signaling pathway in the Zucker rat model of diabetes. The extract also reversed the altered microbiota diversity observed in the diabetic animals by reducing the Firmicutes/Bacteroidetes ratio and increasing the numbers of Proteobacteria and Actinobacteria. In a rat model of dextran sulfate sodium (DSS)-induced colitis, Banskota et al. [89] reported that saffron improved the pathological characteristics of the colon mucosa in association with the reduction of pro-inflammatory cytokines such as tumor necrosis factor alpha (TNF-α) and some interleukins (IL-1β, IL-6). Saffron also reversed the changes in the microbiota caused by the DSS (i.e., depletion of the Proteobacteria phylum and Cyanobacteria) and increased the levels of SCFAs. In contrast, crocetin, one of the main bioactive compounds present in saffron and detected in vivo, has been recently reported to aggravate colitis in a DSS mice model. In this case, crocetin (10 mg/Kg) reduced the diversity and richness of the microbiota detected in the animals treated with DSS [90]. Taken together, all these findings provide preliminary evidence of the potential association between the oral intake of saffron, or of its main compounds crocin and crocetin, and changes in the diversity and composition of the gut microbiota. These studies also suggest that the modulation of the gut microbiota may be one potential main mechanism of action underlying the different health effects attributed to saffron and its main bioactive compounds, including metabolic, inflammatory, and neurological benefits.

### 4.4. Cardiometabolic Benefits of Saffron and Potential Association with Brain Health

In addition to all the above, we should not discard the possibility that the metabolic, inflammatory, and cardiovascular benefits also attributed to saffron may contribute to the neurological effects of this product. Indeed, cardiovascular and metabolic disorders constitute an important risk factor for the development of brain diseases. Obesity, hyperglycemia, hypertension, and dyslipidemia, have all been related to neurological disorders like dementia or depression [91]. In the specific case of type 2 diabetes (T2D), the neurons may be damaged by the increased levels of glucose which might be associated with cognitive deficits, increasing the risk of dementia, and of AD [92]. Several recent clinical trials conducted in T2D patients have reported that the supplementation with saffron can ameliorate metabolic, inflammatory, and oxidative stress biomarkers [26,93,94,95]. In the particular study conducted by Tajaddini et al. [26], intervention with saffron (100 mg/Kg) for eight weeks not only improved the levels of glucose, insulin, triglycerides, and liver function but also reduced depression and improved the quality of sleeping as well as the general quality of life in T2D overweight/obese patients. These results support the hypothesis that one additional potential mechanism underlying the beneficial mental health effects of saffron may be the general improvement of the cardiometabolic status.

## 5. Concluding Remarks and Future Perspectives

In this review we have drawn together the most updated knowledge on: (i) the neurological beneficial effects attributed to saffron and to some of its main bioactive compounds, i.e., crocin and crocetin; (ii) the absorption and metabolism of these compounds, including the potential metabolic influence and interaction with the gut microbiota; and (iii) several proposed physiological routes and mechanisms of action by which the saffron compounds may provide these neurological benefits. A graphic summary of the reviewed information is depicted in Figure 3. Overall and based on the current available evidence, saffron might be a safe therapeutic product with the potential to improve some neurological disorders; nevertheless, there are still several important issues that need to be elucidated [96].

To progress on the understanding of the neurological benefits of saffron and to achieve adequate evidence for real-world recommendations of this product, we next indicate and discuss some of the main limitations identified throughout this review and highlight the main research areas that should be further pursued. Regarding the human clinical studies revised here (Appendix A), some of the main limitations we have observed are common to those already indicated for studies looking at the benefits of other bioactive compounds against other chronic diseases [97], i.e., (i) a rather small sample size (on average ≈ 30 participants per group); (ii) a relatively short duration of the studies (mostly between 28 and 84 d); and (iii) a very small number of dose-response studies. It is thus clear that better-designed and adequately powered intervention trials in larger populations are needed. Given the chronic nature of the neurological disorders and the short duration of both the clinical and pre-clinical studies (hours to days), there is still a lack of knowledge of the long-term effects of saffron and thus longer treatments should also be investigated. Importantly, to identify the potential candidates best responding to the treatment with saffron or saffron compounds, the sample population needs to be better characterized. It has been indicated that there is a large human inter-individual variability in the response to the intake of dietary bioactive compounds. The potential factors that may influence this variability (i.e., sex, age, health status, medication, diet, genetic background, etc.) must be investigated in future studies looking at the effects of the saffron benefits [98].

We also observed a general lack of characterization of the test product. Our study has presented an overview of the complexity and variability of the composition of saffron. In addition to the main bioactive compounds of the saffron spice (crocins, crocetin, safranal), other well-known bioactive compounds (kaempferol, quercetin, anthocyanins, phenolics) are present in the spice and might be also beneficial and contribute to the observed effects. It is important that future research (both clinical and pre-clinical studies) investigating the benefits of saffron must include full characterization of the test product/compound (i.e., total composition, purity, solubility, stability). This information will allow for a better comparison and meta-analysis of different studies and will help to establish the best candidate molecules implicated in the response to saffron. More dose-response studies are additionally needed to establish the most adequate doses of those molecules.

A better definition of the saffron test product will also contribute to improving the elucidation of the absorption, metabolism, and tissue distribution of all the potential metabolites that might be implicated in the response to saffron. Our revision shows that this knowledge is still very limited and focused mainly on the bioavailability of the main saffron compounds, crocin, and crocetin. The identification and quantitation of the potential metabolites is not a trivial task. Published results show a large variability in the detected quantities. For example, in the study conducted by Zhang et al. [49] in Sprague-Dawley rats, the authors detected a plasma Cmax of crocin of 0.06 µg/mL following an oral dose of 60 mg/Kg of pure crocin. However, Shakya et al. [50] provided a 10-times higher dose (600 m/Kg) of pure crocin to the same animal model but reported a Cmax of only 0.0003 µg/mL. A similar response was observed for crocetin. These animal studies already evidence a large variability in the bioavailability of the saffron compounds, not only between studies but also between individuals. This variability was also observed in the human studies [53,54]. Overall, the knowledge of the bioavailability of saffron compounds is still rather limited. It is essential that future animal and human studies conducted with saffron incorporate the qualitative and quantitative identification of the main circulating and tissue metabolites, with special emphasis on those potentially entering the brain tissues. This research should also include the study of the influence of gut microbiota. The action of the microbial enzymes on the saffron compounds may yield new, not yet identified molecules that may be important in a relationship with the neurological effects attributed to saffron. As already stated above, understanding variability, also in the absorption and metabolism of the saffron compounds, will help to understand the responses to saffron and the mechanisms by which saffron or its compounds exerts their effects.

With regards to the current knowledge of the potential mechanisms of action underlying the neurological benefits attributed to saffron, it appears that this product and (or) its components may modulate cell death, oxidative stress, and inflammatory responses in cells and tissues of the nervous system mediated by the regulation of specific key proteins and signaling pathways. However, we found a series of critical issues in the pre-clinical studies used to investigate those mechanisms that poses some concern about the reliability of the proposed results. Overall, the experimental designs applied do not entirely represent the most likely situations in vivo in humans. We found that all the in vitro cell models reviewed here were treated with either saffron extracts, crocin, crocetin or safranal. Of these treatments, the orally administered full saffron extracts will never be as such in contact with neuronal cells. Among the single compounds, the only metabolite so far known to circulate in the plasma of humans is crocetin and thus this compound is a potential candidate to interact with the inner body cells and tissues. Nonetheless, whether crocetin enters the neuronal cells at the high concentrations tested in vitro (µM range) remains to be demonstrated. Future studies should be designed to test the real saffron metabolites, i.e., the molecules/concentration identified in human bioavailability studies, against the specific target cells where those metabolites can be detected.

A number of animal models have also been implemented to investigate the neurological benefits and potential mechanisms of action of saffron. The results reviewed here also support the antioxidant, anti-apoptotic and anti-inflammatory effects of saffron. Of note, these models, as well as most of the in vitro models employed, represent serious diseases such as PD, AD, or ischemia. As already stated, in humans, most clinical trials have been focused on mild disorders like depression, anxiety, or sleeping disorders and there is limited knowledge of the effects of saffron on those more severe neurological conditions. It is then important to explore and compare more adequate models that mimic the specific disorders for which saffron has a demonstrated benefit in humans. Of special interest, the interaction between the saffron compounds and the microbiota appears to be a potential major mechanism of action underlying the neurological effects of saffron through modulation of the gut-brain axis. More studies in this direction are still needed. We additionally noted that most of the animal studies were conducted by dosing the saffron or saffron compounds via i.p. injection. We estimated that to achieve the reported protective effects in humans, we would need to inject equivalent doses (HED) [99] of approximately several hundred to a thousand mg quantities, doses that are most likely impracticable. Once more, the neurological benefits reported in humans are based on the oral administration of the test saffron products and compounds and thus, more animal studies with oral doses that can be translated to human reasonable recommendations are needed. Of note, most studies were conducted on male animals. Since it has been recognized that preclinical research still often ignores sex as a fundamental biological variable and that many neurological disorders show strong sex differences in incidence and disease manifestation [100], it remains essential that future research looking at the neurological benefits of saffron take into consideration this important factor.

In summary, the current evidence supports the beneficial effects of saffron and of its main compounds, crocin, and crocetin, against some neurological disorders. Nevertheless, the knowledge of the balance between the benefits and potential risks of the application of this product, either alone or in combination with other drugs, remains limited. Future clinical trials with many improved designs and more adequate pre-clinical studies are still needed to reinforce the evidence of the therapeutic application of saffron and of its main bioactive components to fight against neurological disorders

## Figures and Tables

**Figure 1 nutrients-14-05368-f001:**
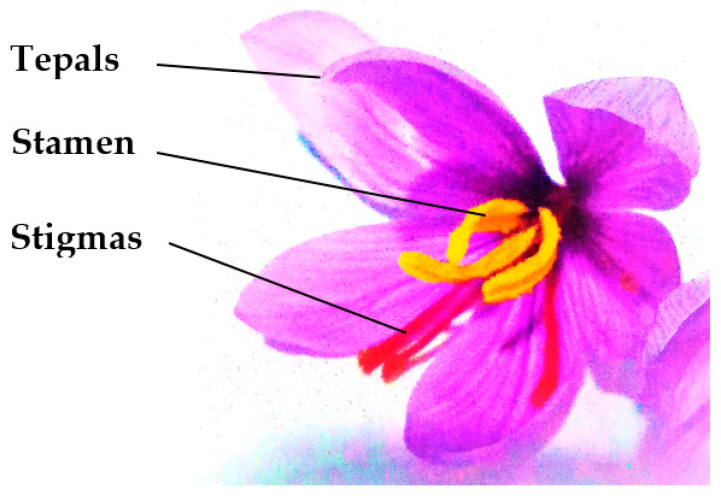
Structure of *Crocus sativus* flower (original authors’ picture).

**Figure 2 nutrients-14-05368-f002:**
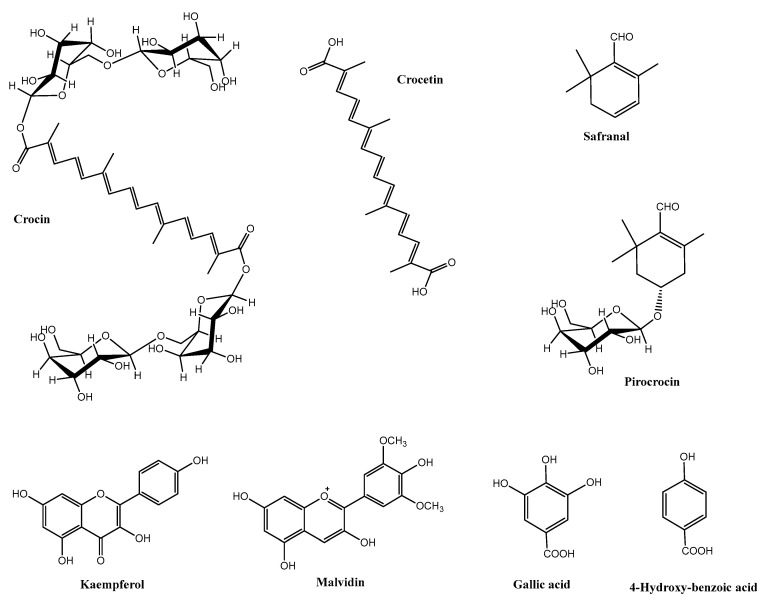
Chemical structure of some of the most representative bioactive compounds present in saffron.

**Figure 3 nutrients-14-05368-f003:**
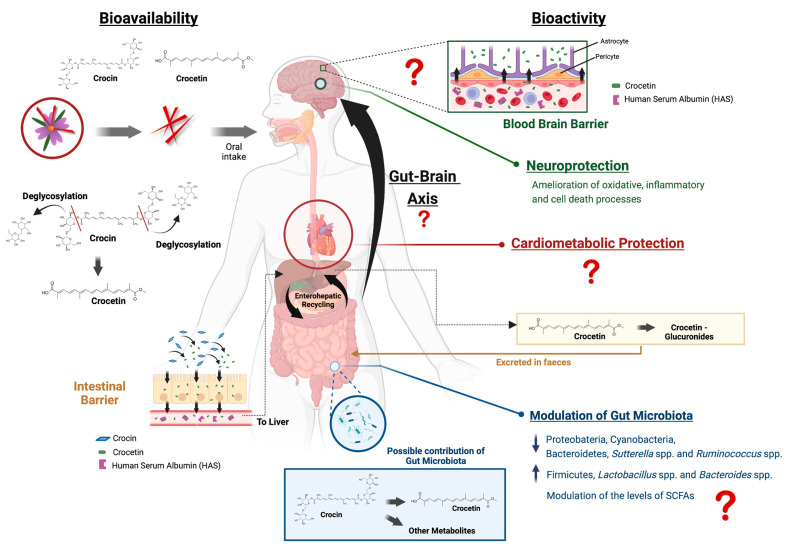
Summary of the current knowledge and hypotheses regarding the bioavailability and potential mechanisms of neuroprotection of saffron and its main bioactive compounds (created with BioRender.com accessed on 22 November 2022).

**Table 1 nutrients-14-05368-t001:** Summary of the main constituents identified in saffron [2,3,4,5,6,7,8,9,10,11,12,20,21].

Component	Saffron Spice (Stigmas)	Saffron Flower Waste (Mostly Tepals)
Primary constituents		
Major nutrients	Carbohydrates, proteins, fats, fiber, reducing sugars	Carbohydrates, proteins, fats, fiber, reducing sugars
Vitamins	Thiamine, riboflavin	Vitamin C
Minerals	K, P, Mg, Ca, Fe, Na	K, P, Mg, Ca, Fe, Na
Other	Amino acids, organic acids	-
Secondary metabolites and bioactive compounds		
Carotenoids	Crocin (*trans*-4-GG, *trans*-3-Gg, crocetin (*trans*- and *cis*- isomers),	*trans*-crocetin, *trans*-crocin, Esters of lutein
Terpenoids	Safranal, pirocrocin	-
Other volatile compounds	Ketones, aldehydes, esters, c13-norisoprenoids, saturated hydrocarbons, acetic acid	Acetoin, butanoic acid, acetic acid, butanal, carbolic acid, phenylethyl alcohol
Flavonoids	Kaempferol, quercetin, epicatechin, anthocyanins	Kaempferol, quercetin, anthocyanins,
Phenolic acids	4-hydroxy benzoic acid, salycilic acid, gallic acid, vanillic acid, rosmarinic acid, chlorogenic acid	Gallic acid, 4-hydroxy-benzoic acid, salicylic acid, chlorogenic acid, hydroxy cinnamic acids, rosmarinic acid
Tannins	Hydrolysable and condensed tannins	Hydrolysable and condensed tannins

**Table 2 nutrients-14-05368-t002:** Pharmacokinetic parameters estimated for crocin and crocetin in the plasma of animal models (rat, mice).

Ref.	Animal Model (N)	Administration Route	Administered Compound	Dose (mg/Kg)	Crocin			Crocetin		
					C_max (µg/mL)_	T_max (min_._)_	AUC_0-tmax (μg·h/mL)_	C_max (µg/mL)_	T_max (min_._)_	AUC_0-tmax (μg·h/mL)_
[47]	ICR mice (8 male)	Intragastric gavage	Crocin (prepared from gardenia fruits)	6.1	-	-	-	0.033	60	-
			Crocetin (prepared by saponification of the Amberlite resin-eluted pigments)	2.1	-	-	-	0.115	60	-
[48]	Sprague-Dawley rats (3 male + 3 female)	Intragastric gavage	Crocin (Nanjing Medical Company, Nanjing, China; 98%)	40	-	-	-	0.830 ± 0.31	60	-
[49]	Sprague-Dawley rats (32 male + 32 female)	Intragastric gavage	Crocin (Chengdu Biopurify Phytochemicals Ltd., Chengdu, China; 99.3%)	29.3	0.050 ± 0.026	114	0.201 ± 0.097	0.688 ± 0.162	120	3.931 ± 0.930
				58.6	0.056 ± 0.034	156	0.310 ± 0.212	1.278 ± 0.623	234	8.669 ± 2.744
				117.2	0.082 ± 0.039	114	0.394 ± 0.169	1.249 ± 0.788	246	10.61 ± 8.207
			Crocetin (Chengdu Biopurify Phytochemicals Ltd.; 96.6%)	9.8	ND	ND	ND	0.387 ± 0.228	162	3.062 ± 1.607
				19.7	ND	ND	ND	0.572 ± 0.380	174	4.587 ± 3.009
				39.3	ND	ND	ND	0.658 ± 0.316	150	5.258 ± 2.594
[50]	Sprague-Dawley rats (Control—5 male)	Intragastric gavage	Crocin (Sigma, St. Louis, MO, USA)	600	3.04 × 10^−4^ ± 1.53 × 10^−4^	168 ± 66	1.25 × 10^−3^ ± 6.09 × 10^−4^	1.03 × 10^−2^ ± 2.00 × 10^−3^	204 ± 54	6.79 × 10^−2^ ± 7.99 × 10^−3^
	Sprague-Dawley rats (Antibiotic pre-treatment—5 male)				7.49 × 10^−5^ ± 3.01 × 10^−5^	150 ± 84	4.71 × 10^−4^ ± 4.48 × 10^−4^	9.73 × 10^−3^ ± 2.37 × 10^−3^	132 ± 24	3.51 × 10^−2^ ± 4.14 × 10^−3^

ICR: Institute of Cancer Research; AUC: area under curve; C_max_: plasma maximum concentration; T_max_: time needed to achieve maximum concentration; ND: not detected.

**Table 3 nutrients-14-05368-t003:** Pharmacokinetic parameters estimated for crocetin in human plasma.

Ref.	Population (N, Gender, Age)	Product	Dose	Crocetin		
				C_max (µg/mL)_	T_max (min_._)_	AUC_0-tmax (μg·h/mL)_
[53]	4 healthy volunteers (1 male + 3 female; 25 to 35 years)	Saffron Infusion (trans-crocin-4: 59.5 ± 0.2%; trans-crocin-3: 21.0 ± 0.4%; cis-crocin-4: 5.0 ± 0.1%; trans-crocin-2: 1.3 ± 0.2%; cis-crocin-3: 8.2 ± 0.2%; cis-crocin-1: 1.1 ± 0.3%)	200 mg stigmas/150 mL hot water	0.41–1.21	120 min	NA
[54]	13 healthy volunteers (5 male + 8 female; 18 to 30 years)	affron^®^ (Saffron Extract) (safranal: 0.04 ± 0.01%; picrocrocin: 3.21 ± 0.07%; kaempferol diglucoside: 0.13 ± 0.01%; crocins: 3.63 ± 0.05%; crocetin: 0.03 ± 0.01%)	4 tablets—56 mg	0.26 ± 0.12	60 min	21.07
			6 tablets—84 mg	0.39 ± 0.10	90 min	26.15

AUC: area under curve; C_max_: plasma maximum concentration; T_max_: time needed to achieve maximum concentration; NA: not available.

**Table 4 nutrients-14-05368-t004:** In vitro studies looking at the effects of saffron and its main bioactive constituents against neurological disorders.

	In Vitro Model		Saffron Treatment			Reported Effects
Ref.	Description	Biological Role in the Brain or Nervous System	Test Compound Purity	Concentration, Time of Exposure	Reported Mechanisms	Potential Neurological Benefit
	Cell culture model					
[62]	Mouse hippocampal neuronal cell line HT-22 treated with OGD to induce ischemia	Hippocampal neurons play a major role in the functioning of the human brain and in the memory	Crocin (Nanjing Jingzhu Biotechnology Co. Ltd., Nanjing, China) 98%	1 µg/mL, 2 µg/mL and 5 µg/mL, 14 h	Improves cell viability, decreases apoptosis, decreases ROS, increases the expression of p-PI3K, p-Akt, and p-mTOR (activation of the PI3K/mTOR pathway) while decreases the expression of LC-3 II/I and BECN1	Neuroprotective effects against cerebral infarction by inhibiting autophagy and reducing oxidative stress
[63]	Microglia BV2 cells isolated from a 10-days old female mouse treated with neurotoxin MPTP to induce PD like symptoms	Brain macrophages. Regulate brain development, maintenance of neuronal networks, and injury repair. Protect against infection and inflammation	Crocetin dialdehyde (Sigma, St. Louis, MI, USA) >95%	2.5 μM, 5 μM, 10 μM, 24 h	Decreases the expression of inflammatory associated genes (p-p65 and pro-/Cleaved Casp1) and cytokines (IL-1β, IL-6, IL-10, TNF-α, inducible nitric oxide synthase, COX2). Improves mitochondrial function by blocking intracellular reactive oxygen species (ROS) levels, increasing mitochondrial membrane potential and reducing the content of calcium in the cytosol	Inhibition of inflammatory conditions in the brain and amelioration of PD associated symptoms
[64]	PC-12 cells isolated from a rat male exposed to OGD to induce ischemia	PC-12 cells, isolated from adrenal medulla pheochromocytoma, mixture of neuroblastic cells and eosinophilic cells. Used to study biological functions of neuronal cells and brain diseases	Safranal (Sigma) ≥90%	10–160 μM 2 h	Attenuation of oxidative brain injury via reducing intracellular ROS levels, oxidative DNA damage and cell apoptosis at 40 μM and 160 μM of safranal	Neuroprotection due to the modulation of oxidative stress and apoptotic responses in the brain
[66]	PC-12 cells isolated from a rat male (brain ischemia)	PC-12 cells, isolated from adrenal medulla pheochromocytoma of rats, mixture of neuroblastic cells and eosinophilic cells. Used to study biological functions of neuronal cells and brain diseases	Crocin NI	1 µM, 10 μM, NI	Prevention of neuronal cell apoptosis, reduction of ROS generation, increased cellular levels of GSH by the activation of GR and restored SOD activity, especially at 10 μM of crocetin	Potential neuroprotection due to the antioxidant activity: reducing oxidative stress and neuronal apoptosis
[67]	PC-12 cell culture model isolated from a rat male treated with neurotoxin MPP+ to induce PD like symptoms	PC-12 cells, isolated from adrenal medulla pheochromocytoma of rats, mixture of neuroblastic cells and eosinophilic cells. Used to study biological functions of neuronal cells and brain diseases	Crocin Crocetin (Sigma) ≥95% ≥95%	0.1 μM, 1 μM, 10 μM or 100 μM, 0, 3, 6 or 9 h after MPP+ exposure	Improvement of mitochondrial function and protection against cell apoptosis at 10 and 100 μM of crocin	Protection of cytotoxicity by inhibiting the activation of pro-apoptotic factor Casp12 by its anti-apoptotic activity
[65]	Primary dopaminergic cells isolated from rat embryos treated with rotenone to induce PD like symptoms	Dopaminergic cells are collections of neurons in the central nervous system that synthesize the neurotransmitter dopamine. The loss of these neurons is associated with neurological disorders such as PD	Safranal (Sigma) ≥ 90% ≥90% ≥90%	10 μM, 15 μM, 20 μM, and 50 μM, 4 h	Suppression of ROS generation and cell apoptosis by the inhibition of Keap1 protein expression and promotion of nuclear translocation of Nrf2, especially at 50 μM of safranal	Neuroprotection against oxidative stress via the modulation of the Nrf2 signalling pathway that is involved in the protection of PD
[70]	Human neuronal cell line SH-SY5Y from a female neuroblastome subjected to α-Syn toxicity	Representative of human nervous cells	Saffron (Matsuura Yakugyo; Aichi, Japan) in DMSO	6 µg/mL and 20 µg/mL	Saffron restores cell viability at the highest concentration	Cytoprotective effects in human cells
	α-Syn PFF model					
[68]	Solution of α-Syn fibrils	α-Syn is a 140-amino acid protein abundant in the human brain. Its abnormal aggregation and accumulation produces neurodegenerative diseases such as PD	Saffron powder (Matsuura Yakugyo, Japan); Crocin (Carbosynth & Adooq Bioscience, Irvine, CA, USA); Crocetin, safranal (Toronto Research Chemicals, North York, ON, Canada) NI	αS aggregation: 0.5 μL of saffron compounds, 3 h at 37 °C αS dissociation: 0.5 μL of saffron compounds, 15 min at 37 °C	Inhibition of α-Syn aggregation and promotion of the dissociation of α-Syn fibrils. Crocetin was the most potent bioactive compound	Effective neuroprotection to prevent and treat neurological disorders generated by abnormal α-Syn aggregation
[69]	Solution of E46K mutant forms of α-Syn fibrils	The E46K mutation of the α-Syn gene is linked to cause PD	Crocin (Booali Research Center, Mashhad, Iran) NI	Different molar ratios α-synuclein:crocin (1:0, 1:0.25, 1:0.5), different times	Inhibition of amyloid fibril formation in E46K α-Syn, interacting with its hydrophobic surface area	Potential therapeutic neuroprotection on the fibrillation pathway, inhibiting the dimerization and polymerization of α-Syn fibrils

DMSO: dimethyl sulfoxide; MPTP: 1-methyl-4-phenyl-1,2,3,6-tetrahydropyridine; OGD: oxygen-glucose deprivation; MPP+: methyl-4-phenylpyridinium; PD: Parkinson’s disease; PFF: preformed fibrils; ROS: reactive oxygen species; p-p65: phosphorylated protein p65 (DNA binding transcription factor); IL: interleukin; TNF-α: tumor necrosis factor-α; iNOS: inducible nitric oxide synthase; COX2: cyclooxygenase-2; GSH: glutathione; GR: glutathion reductase; Casp: caspase; Keap1: Kelch-like ECH-associated protein; Nrf2: nuclear factor erythroid 2-related factor 2; p-PI3K: phosphorylated phosphoinositide 3-kinase; p-Akt: phosphorylated AKT serine/threonine kinase 1; p-mTOR: phosphorylated mechanistic target of rapamycin kinase; LC-3 II/I: microtubule-associated proteins 1A/1B light chain 3A; BECN1: beclin-1; α-Syn: NI: not indicated.

**Table 5 nutrients-14-05368-t005:** Animal studies looking at the effects of saffron and its main bioactive constituents against neurological disorders.

		Saffron Treatment			Reported Effects	
Ref.	Animal Model Sex	Test Compound Purity	Dose Administration Route Duration of Treatment	HED (mg/60 Kg Person)	Reported Mechanisms	Potential Neuroprotection
[70]	*Drosophila* model of PD overexpressing several human mutants α-Syn in their neurons or eyes Male	Saffron (Matsuura Yakugyo, Japan) suspended in distilled water; Crocetin extracted from saffron NI	Saffron: 3, 10 and 30 µg/mL; crocetin: 0.3, 1 and 3 µg/mL, p.o. (added into the feed) ≈80 days	ND	Improvement of the climbing ability, life expand, and eye phenotype but the results were α-Syn mutation-dependent. The results were especially seen at the higher concentrations of saffron and crocetin	Potential protective effect against PD progression by interfering with α-Syn aggregation process
[73]	Wistar rats (3-weeks-old) induced brain I/R injury Male	Saffron stigmas (from a local market) 300 g/3 L of 80% ethanol during 3 days, filtered and concentrated	100 and 200 mg/kg b.w. i.p. 21 days prior to brain I/R injury and 4 days during I/R injury	≈975 mg and ≈1950 mg	Attenuation of lipid peroxidation, decrease of NO and BNP, reversal of the depletion of GSH, upregulation of the expression of VEGF and decrease of the expression of CASP3 and Bax in brain tissue at both concentrations of saffron	Potential neuroprotection through the reduction of oxidative stress and apoptosis
[74]	Intracerebral hemorrhage mouse model Male	Crocin (Sigma) ≥95%	40 mg/Kg b.w. i.p. 2 h after intracerebral hemorrhage	≈195.0 mg	Improvement of brain edema and neurological defficient score. Increased SOD and GSH-Px activity while reduction of MDA. Inhibition of ferroptosis of neurons and molecular regulation of the expression of GPX4, FTH1, SLC7A11. Translocation of Nrf2.	Alleviation of intracerbral hemorrhage
[75]	C57BL/6J mice hypoxia-ischemia-induced NI	Crocin (Sigma) ≥95%	10 mg/Kg b.w. i.p. every 12 h during 48 h	≈50.0 mg	Reduction of MDA, NO and ROS in the brain and inhibition the expression of pro-inflammatory mediators (combined effect with hypothermia)	Improvement of the neurological function and neuroprotection by the reduction of the oxidative stress and inflammation in brain
[76]	Adult Wistar Albino rats Aβ1–42 AD-induced Male	Crocin powder dissolved in saline solution NI	30 mg/kg b.w. i.p. once a day during 12 days	≈300 mg	Improvement of synaptic loss and reduction of neuronal cell apoptosis	Protection of neuronal cell death which is responsible for cognitive impairment and dementia in AD
[77]	Neonatal Sprague Dawley rats treated with MK-801 to induce schizophrenia-like symptoms NI	Crocin (Chengdu Puri Technology Limited, Chengdu, China) dissolved in sterile saline solution ≥97%	25 and 50 mg/kg b.w. i.p. once a day, on postnatal day 7 to day 14 (during 7 days)	≈250 mg and ≈500 mg	Amelioration of schizophrenia symptoms via regulation of SIRT1 and downstream BDNF proteins expression in the hippocampus. Relief of the oxidative stress by reducing MDA levels and increasing the activity of CAT, GPx, and SOD in the hippocampus	Neuroprotection effects since SIRT1 participate in the pathogenesis of neurodegenerative disorders, playing important roles in several biological processes
[78]	CFT-Swiss mice (8-week old) treated with rotenone to induce PD like symptoms Male	Crocin (Sigma): 10 mg/mL prepared in double distilled water ≥95%	25 mg/kg b.w. i.p. once a day during 7 days	≈120 mg	Restoration of the levels of dopamine and α-Syn and enhancement of antioxidant enzymes and mitochondrial enzymes function in the striatal brain region	Neuroprotection by reducing the oxidative stress and improving mitochondrial function which is necessary for the high energy demand of the brain
[80]	Wild type C57BL/6J mice, aged 2–12 months Male and female	*trans*-crocin-4 isolated from *Crocus sativus* stigmas and dissolved in saline solution >90%	50 mg/kg and 150 mg/kg b.w. i.p. duration NI	≈250 mg and ≈750 mg	Changes in the metabolome, detection of some metabolites with important roles in neuroprotection (21-Hydroxypregnenolone and 17α-epiestriol participating in the steroid biosynthesis pathway; glutaryl carnitine and riboflavin related with the oxidative stress, and betaine involved in inhibitory neurotransmitter production pattways)	Potential prevention of neurological disorders trough changes in the metabolomic profile
[63]	C57BL/6J mice treated with MPTP to induce PD like symptoms Male	Crocetin dialdehyde (Sigma) ≥95%	50 and 100 mg/Kg (NI per b.w. or per kg of feed) p.o. once a day during 11 days	≈250 mg and 500 mg	Attenuation of the expression of the TH marker in the striatal brain regions at both concentrations of crocetin	Protection of dopaminergic neurons against toxin-induced damage
[79]	Adult Wistar rats of transient focal cerebral ischemia induced by MCAO Male	Safranal (Sigma) ≥90%	72.5 and 145 mg/kg b.w. i.p. 0, 3, and 6 h after ischemia/reperfusion injury	≈700 mg and ≈1400 mg	Inhibition of the oxidative stress in the brain tissues, reducing MDA levels and increasing the total sulfhydryl content at both concentrations of safranal	Potential neuroprotective effects by modulating the oxidative stress

NI: not indicated; ND: not determined; b.w.: body weight; p.o.: per oral; i.p.: intraperitoneal injection; AD: Alzheimer’s disease; PD: Parkinson’s disease; α-Syn: α-Synuclein; MPTP:1-methyl-4-phenyl-1,2,3,6-tetrahydropyridine; TH: tyrosine hydroxylase; MK-801 maleate: (Dizocilpine), non-competitive NMDA (N-metil-D-aspartato) antagonist; I/R: ischemia/reperfusion injury; MCAO: middle cerebral artery occlusion; MDA: malondialdehyde; ROS: reactive oxygen species; CAT: catalase; CASP: caspase; Bax: BCL2 associated X, apoptosis regulator; NO: nitric oxide; FTH1: ferritin heavy chain 1; SLC7A11: solute carrier family 7 member 11; Nrf2: nuclear factor erythroid 2-related factor 2; SOD: superoxide dismutase; GSH: glutathion; GSH-Px: glutathion peroxidase activity; GPX: glutathion peroxidase; VEGF: vascular-endothelial growth factor; SIRT1: silent information regulator-1; SOD: superoxide dismutase; BNP: brain natriuretic peptide; BDNF: brain derived neurotrophic factor; HED: human equivalent dose (mg for a 60 Kg person). HED was calculated using the following equation: HED (mg/kg) = animal dose (mg/kg) × Km. Km: correction factor ratio estimated for each species (Km = 0.162; for mice Km = 0.081).

## Data Availability

All the data used in this review are included in the Tables and in the Appendix A of the manuscript.

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
