# Peer review of "Saffron against Neuro-Cognitive Disorders: An Overview of Its Main Bioactive Compounds, Their Metabolic Fate and Potential Mechanisms of Neurological Protection"

_nutrients, 2022, doi:10.3390/nu14245368_

Round 1

Reviewer 1 Report

Dear authors,

Correct the manuscript as indicated in the text.

Author Response

Uploaded as a pdf

Reviewer 2 Report

I think this is a comprehensive and high quality review on the chemical composition, metabolic transformation, neuroprotective pharmacological activity and clinical progress of saffron. I agree to receive and publish it. I believe it will be helpful to the correct understanding of similar widely used natural drugs (products), but at the same time I have a little suggestions:

1.P 4,line 144,metanalyses should be meta analyses.

2.Table 2, ICR: Intestinal cancer research, should be ICR:Institute of Cancer Research.

3. Line 418 and line 414, amyloid-β peptide,the  abbreviation should be Aβ.

4.Although the main purpose of the full text is to discuss the activity, we know that any drug has both benefits and risks, because it has a relatively good activity, whether there will be some precautions or adverse reactions in drug combination, and it is suggested that it would be more reasonable and comprehensive to properly mention it in the discussion section.

Author Response

Uploaded as a pdf
